# Elastography Techniques for the Assessment of Liver Fibrosis in Non-Alcoholic Fatty Liver Disease

**DOI:** 10.3390/ijms21114039

**Published:** 2020-06-05

**Authors:** Yasushi Honda, Masato Yoneda, Kento Imajo, Atsushi Nakajima

**Affiliations:** Department of Gastroenterology and Hepatology, Yokohama City University Graduate School of Medicine, Yokohama 236-0004, Japan; y-honda@umin.ac.jp (Y.H.); dryoneda@yahoo.co.jp (M.Y.); kento318@yokohama-cu.ac.jp (K.I.)

**Keywords:** elastography, non-alcoholic fatty liver disease, vibration-controlled transient elastography, point shear wave elastography, two-dimensional shear wave elastography, magnetic resonance elastography

## Abstract

Non-alcoholic fatty liver disease (NAFLD) is expected to increase in prevalence because of the ongoing epidemics of obesity and diabetes, and it has become a major cause of chronic liver disease worldwide. Liver fibrosis is associated with long-term outcomes in patients with NAFLD. Liver biopsy is recommended as the gold standard method for the staging of liver fibrosis. However, it has several problems. Therefore, simple and noninvasive methods for the diagnosis and staging of liver fibrosis are urgently needed in place of biopsy. This review discusses recent studies of elastography techniques (vibration-controlled transient elastography, point shear wave elastography, two-dimensional shear wave elastography, and magnetic resonance elastography) that can be used for the assessment of liver fibrosis in patients with NAFLD.

## 1. Introduction

Non-alcoholic fatty liver disease (NAFLD) has become a major cause of chronic liver disease worldwide. The prevalence of NAFLD is currently estimated to be 25% in the general population [1,2], 90% of people with obesity, and 60% of patients with type 2 diabetes mellitus [3,4,5]. Furthermore, the prevalence of NAFLD is expected to increase because of the epidemics of obesity and diabetes [6]. Liver fibrosis, but not non-alcoholic steatohepatitis, is associated with long-term outcomes in patients with NAFLD [7,8]. Liver fibrosis is categorized into non-fibrosis (stage 0), mild fibrosis (stage 1), significant fibrosis (stage 2), advanced fibrosis (stage 3), and cirrhosis (stage 4) (Figure 1) [9]. Liver biopsy is conventionally recommended as the gold standard method for the diagnosis of NAFLD and the staging of liver fibrosis [10]. However, it has several problems, such as sampling error, inter- and intra-observer variation, a risk of complications, and high cost [11,12]. Therefore, simpler and noninvasive methods for the assessment of liver fibrosis have been explored.

Conventional ultrasonography (US), computed tomography, and magnetic resonance imaging (MRI) are useful for the diagnosis of chronic liver disease and cirrhosis and the detection of hepatocellular carcinoma. However, these imaging methods cannot accurately differentiate the various stages of liver fibrosis. Conversely, elastography techniques using US or MRI are performed to measure liver stiffness, which increases in the presence of fibrosis. Therefore, during the last two decades, elastography techniques have been developed as quantitative noninvasive methods for the assessment of liver fibrosis that can be used in place of liver biopsy. Several US-based elastography techniques have been developed, the most important of which is shear wave elastography, which can be divided into vibration-controlled transient elastography (VCTE), point shear wave elastography (pSWE), and two-dimensional shear wave elastography (2D-SWE). In this review, we discuss recent studies of the use of elastography techniques for the assessment of liver fibrosis in NAFLD. Figure 2 shows illustrations and images of each elastography technique, and Table 1 lists the meta-analyses of studies of elastography techniques for the diagnosis of liver fibrosis in patients with NAFLD that have been conducted to date. Appendix A lists 47 studies included in the meta-analyses.

## 2. Vibration-Controlled Transient Elastography

VCTE was developed in 1992 as the first US-based elastography technique [13] and requires a one-dimensional probe and an ultrasonic transducer. The probe is placed over an intercostal space, and a low-amplitude 50-Hz mechanical pulse is then transmitted from the probe to the liver, inducing the propagation of an elastic shear wave through the tissue. The propagation velocity that is measured by VCTE is positively related to the liver stiffness, within the range of 1.5 to 75 kPa [14]. VCTE assesses a volume of liver around 100 to 200 times the size of a liver biopsy. Three different probes can be used to make measurements under various circumstances: A standard M probe (3.5 MHz) is used for adults, an XL probe (2.5 MHz) is used for overweight patients, and an S probe (5.0 MHz) is used for children. Lower-frequency probes are suitable to reduce wave attenuation in patients who have a high degree of abdominal adiposity or a long distance between the skin and liver surface [15].

Yoneda et al. [16] first reported the usefulness of VCTE for estimating the severity of liver fibrosis in patients with NAFLD in 2007. VCTE performed using the FibroScan (Echosens, Paris, France) became the first Food and Drug Administration-approved US-based elastography technique in 2013. VCTE was also included in the European Association for the Study of the Liver Clinical Practice Guidelines for the assessment of liver fibrosis in patients with chronic hepatitis B and C virus infection [17]. Although VCTE is a blind technique, it is the most commonly used technique for screening, treatment monitoring, and longitudinal follow-up in various chronic liver diseases, and is the best-validated elastography modality worldwide. In patients with NAFLD, repeated measurements of liver stiffness made using VCTE are useful for long-term monitoring and the prediction of liver-related complications and cardiovascular events [18,19]. In addition, repeated measurements of liver stiffness can reduce false-positive diagnosis of advanced fibrosis in patients with NAFLD [20,21].

The benefits of VCTE are its extensive validation, availability, and high patient acceptance. In addition, there is good intra- and inter-observer reproducibility (intra-class correlation coefficient (ICC) = 0.98) in patients with various liver diseases, including NAFLD [22]. However, VCTE has some technical and patient-related limitations: The equipment requires recalibration every 6 to 12 months to ensure technical stability [15], and the failure rate of VCTE using a standard M probe is high (6.7%–29.2%). Common patient-related limitations are obesity, operator inexperience, narrow intercostal spaces, acute inflammation, and ascites [23,24,25,26].

Kwok et al. [27] performed the first systematic review and meta-analysis of studies of VCTE (M-probe). The meta-analysis included 854 patients with NAFLD and showed that the sensitivity and specificity for the diagnosis of stages 2, 3, and 4 fibrosis were 0.79 and 0.75, 0.85 and 0.85, and 0.92 and 0.92, respectively. In the most recent meta-analysis of VCTE (M-probe), which was based on 11 studies and 1753 patients with NAFLD, the area under the receiver operating characteristic curve (AUROC) for the diagnosis of stages 2, 3, and 4 fibrosis was 0.85, 0.92, and 0.94, respectively [28]. The authors concluded that VCTE is useful for the staging of liver fibrosis in patients with NAFLD, particularly for those with advanced fibrosis and cirrhosis. Xiao et al. [29] performed a meta-analysis of the use of the XL probe. The meta-analysis included three studies involving 318 patients with NAFLD and showed that the AUROC for the diagnosis of stages 2, 3, and 4 fibrosis was 0.82, 0.86, and 0.94, respectively.

## 3. Point Shear Wave Elastography

The pSWE was originally developed by Siemens (Erlangen, Germany). Unlike VCTE, which uses a mechanical impulse, pSWE uses an acoustic radiation force impulse to induce shear waves in liver tissue. The performance of pSWE is assisted by its incorporation into standard B-mode US acquisition. This technique permits an operator to visualize the liver tissue and select a region without blood vessels, rib shadows, or bile ducts. The pSWE produces a single point of energy within the liver and targets a region of interest (ROI) of 5 × 10 mm using B-mode imaging [30]. The measured shear wave speed is expressed in m/s and converted to Young’s modulus in kPa for the estimation of tissue stiffness. The pSWE has shown high levels of repeatability and reproducibility in studies involving same-day comparisons [31]. The other advantages of pSWE include high intra-observer (ICC = 0.89–0.90) and inter-observer (ICC = 0.81–0.85) coefficients, which were calculated in a cohort of patients with various liver diseases including NAFLD [31,32,33]. Although pSWE may also overestimate the severity of liver fibrosis in patients with acute inflammation, it is not limited by the presence of ascites, unlike VCTE. Indeed, the failure rate of pSWE in healthy volunteers is low (1%–2%) [34].

The systematic review and meta-analysis by Liu et al. [35] showed that pSWE had a modest level of accuracy for the detection of stage 2 fibrosis (summary sensitivity, 0.80; summary specificity, 0.85; and AUROC, 0.90). Jiang et al. [28] performed a meta-analysis of studies of pSWE in patients with NAFLD and found that the AUROC for the diagnosis of stages 2, 3, and 4 fibrosis was 0.86, 0.94, and 0.95, respectively. The authors concluded that pSWE is also useful for the staging of liver fibrosis, particularly in patients with advanced fibrosis or cirrhosis. The most recent systematic review and meta-analysis, published in 2020, was based on 13 studies and involved 1147 patients with NAFLD [36]. This meta-analysis showed that the AUROC for the diagnosis of stages 2, 3, and 4 fibrosis was 0.89, 0.94, and 0.94, respectively.

## 4. Two-Dimensional Shear Wave Elastography

Two-dimensional SWE was first introduced on a diagnostic imaging device called the Aixplorer (SuperSonic Imagine, Aix-en-Provence, France) [37]. It also uses acoustic radiation force impulse and is now available on US scanners produced by most major manufacturers. Two-dimensional SWE involves the focusing of acoustic energy to multiple sites in the liver and generates real-time and 2D quantitative maps of liver tissue elasticity using standard B-mode US imaging over a significantly larger area of tissue (35 × 25 mm) than VCTE and pSWE [38,39,40]. Stiffer tissues appear red and softer tissues appear blue on the display [37,40]. The mean shear wave speed (m/s) is derived from multiple measurements obtained from tissue within the ROI, which can be adjusted in terms of size and location. The measured shear wave speed can be algebraically converted to Young’s modulus (kPa). The advantages of 2D-SWE, as well as pSWE, are its rapidity, patient acceptance, and high intra-observer (ICC = 0.93–0.95) and inter-observer (ICC = 0.88) coefficients [41]. In addition, the operator can explore a large field of view by choosing an ROI. However, it is necessary to standardize the selection of an appropriate ROI. The sampling time associated with 2D-SWE may be longer than those associated with VCTE or pSWE because larger tissue volumes are assessed. In one study, the estimated failure rate of 2D-SWE was about 5% in 79 patients (25 healthy patients, 26 with various liver diseases) [42]. To date, no meta-analyses of studies of 2D-SWE in patients with NAFLD have been conducted. Therefore, the diagnostic accuracy of 2D-SWE in patients with NAFLD requires further investigation.

## 5. Magnetic Resonance Elastography

Magnetic resonance elastography (MRE) was developed at the Mayo Clinic in 1995 [43], introduced into clinical practice in 2007, and approved by the Food and Drug Administration in 2009. It is an MRI-based technique for the quantitative imaging of tissue stiffness and is currently the most accurate noninvasive imaging method for the diagnosis of liver fibrosis [44,45,46,47]. Today, MRE is available on MR scanners made by three of the principal manufacturers (General Electric, Milwaukee, WI, USA; Philips Medical Systems, Best, Netherlands; and Siemens Healthineers, Erlangen, Germany) at 1.5 T and 3 T field strengths.

Quantitative stiffness images (elastograms) of the liver can be rapidly obtained during breath-holding and can, therefore, be readily included in conventional liver MRI protocols [48]. The liver volume that is measurable using MRE is typically ≥250 mL and up to one-third of the liver volume [45,49,50]. A more advanced version of three-dimensional MRE can evaluate the entire liver volume and was used in a recent prospective study [51]. Therefore, MRE can be used to assess the entire liver with a high success rate [52]. Furthermore, unlike US-based techniques, the success of MRE is operator-independent [47] and is minimally affected by obesity, ascites, and bowel interposition between the liver and abdominal wall [44]. MRE is also highly repeatable, and there is high inter-observer and intra-observer reproducibility among the scanner models [53,54,55,56]. The estimated failure rate of MRE is about 5% in patients with various liver diseases [57], and substantial iron deposition in the liver is the most common cause of failure. However, patients who are claustrophobic and have undergone implantation of MR-incompatible devices cannot tolerate MR examinations. Motion artifacts such as cardiac impulses are another cause of failure because MRE is a motion-sensitive technique. MRE should be conducted after ≥4 h of fasting because liver stiffness measurements may increase due to postprandial portal blood flow [58].

A systematic review and analysis of pooled individual participant data by Singh et al. [59] showed that the optimal cut-off MRE value for the diagnosis of stages 1, 2, 3, and 4 fibrosis in patients with NAFLD was 2.88, 3.54, 3.77, and 4.09 kPa, respectively. The authors also calculated the AUROC for the diagnosis of each of these stages (0.86, 0.87, 0.90, and 0.91, respectively). In the most recent meta-analysis by Xiao et al. [29] of five studies involving 628 patients with NAFLD, the AUROC for the diagnosis of stages 2, 3, and 4 fibrosis using MRE was 0.88, 0.93, and 0.92, respectively. The authors concluded that MRE may have the highest diagnostic accuracy for the staging of liver fibrosis.

## 6. Comparison of Elastography Techniques in Patients with Non-Alcoholic Fatty Liver Disease

Table 2 outlines the advantages and limitations of the above-described elastography techniques. While US-based elastography techniques are relatively inexpensive and simple, they are limited by a high technical failure rate in patients with obesity [24,60]. In addition, these techniques evaluate only a limited volume of the liver, and the results may be influenced by inflammation, cholestasis, and hepatic congestion [61]. The results of liver stiffness by MRE may also be influenced by cholestasis and hepatic congestion [62]. Jiang et al. [28] compared the technical failure ratios of pSWE and VCTE in patients with NAFLD in a meta-analysis and found that the proportion of failed measurements was >10-fold greater when VCTE (M-probe) was used (VCTE: 11.3% (187/1649) and pSWE: 0.8% (6/733)). Several studies have shown that the success ratio of VCTE is much lower when it is performed on patients with a high body mass index of >25 kg/m^2^ [63,64]. However, obesity seems to have less influence on pSWE. Nevertheless, the accuracy of pSWE is affected by the presence of severe steatosis [65,66]. Cassinotto et al. [39] reported that liver stiffness measurements made using pSWE are unreliable in 18.2% of patients with NAFLD. As with VCTE and pSWE, technical failures of 2D-SWE occur in patients with obesity and in those with a thick subcutaneous fat layer [39], and most patients with NAFLD are obese and have thick subcutaneous fat layers. To circumvent this problem, the XL probe was developed for use in overweight patients. A recent prospective study showed no significant difference in diagnostic accuracy between M and XL probes using probe-specific cut-off values [67]. If the choice of the M probe or the XL probe is made using the automatic probe recommendation tool of the VCTE device, the applicability of VCTE increases to 97% in patients with NAFLD [68,69]. However, there has been insufficient validation of the use of the XL probe. Although MRE is expensive and not widely available, its use for the measurement of liver stiffness is not associated with similar confounders. A systematic review and analysis of pooled individual participant data by Singh et al. [59] showed that the diagnostic performance of MRE is robust, stable, and independent of sex, obesity, and inflammation. Imajo et al. [70] found that although the costs associated with MRE are higher than those associated with VCTE, MRE has the advantage of yielding data for the entire liver, which is useful for screening for other diseases, including hepatocellular carcinoma.

MRE has many advantages over US-based elastography techniques for the evaluation of liver fibrosis. In 2017, Xiao et al. [29] conducted a systematic review and meta-analysis of 64 articles involving 13,046 patients with NAFLD to compare the diagnostic performance of noninvasive indexes (aspartate aminotransferase-to-platelet ratio index, fibrosis-4 index, BARD score, NAFLD fibrosis score, VCTE (M- and XL-probe), SWE, and MRE for the prediction of significant fibrosis, advanced fibrosis, and cirrhosis. The authors found that MRE offered the best diagnostic performance for the staging of liver fibrosis. Other studies have also demonstrated that MRE is superior to VCTE and noninvasive indexes for the diagnosis of liver fibrosis in patients with NAFLD [51,71,72,73]. Because MRE has the highest accuracy for the diagnosis of liver fibrosis, it is being increasingly regarded as a promising surrogate measurement for the monitoring of disease progression and therapeutic endpoints [74]. The most recent prospective cohort study by Ajmera et al. [75] investigated the clinical utility of an increase of MRE in predicting fibrosis progression in patients with NAFLD with paired biopsies and paired MRE measurements. The authors reported that a 15% increase in MRE was associated with histologic fibrosis progression.

## 7. Conclusions

Although each elastography technique has its advantages and limitations, VCTE and MRE are considered the methods of choice. According to the clinical practice guidelines published by the European Association for the Study of the Liver, VCTE is an acceptable noninvasive procedure for the identification of patients at low risk of advanced fibrosis/cirrhosis [76]. In addition, according to practice guidance published by the American Association for the Study of Liver Diseases, VCTE and MRE are clinically useful tools for the identification of advanced fibrosis in patients with NAFLD [77]. However, pSWE and 2D-SWE are not recommended in the current guidelines for NAFLD. One reason is that there are no data for follow-up using pSWE and 2D-SWE in patients with NAFLD. In addition, VCTE and MRE have the advantage of being used to evaluate not only liver fibrosis but also liver steatosis. The controlled attenuation parameter, which is based on the properties of the ultrasonic signals acquired using VCTE, is a novel means of grading steatosis by measuring the degree of ultrasound attenuation by hepatic steatosis [78]. The FibroScan-AST (FAST) score, which combines the liver stiffness measurement and the controlled attenuation parameter measured by VCTE and aspartate aminotransferase, was recently proposed [79]. This score can identify patients with non-alcoholic steatohepatitis (NAFLD activity score of ≥4 and fibrosis stage ≥2) and has been validated in large global cohorts. Furthermore, the proton density fat fraction is an MRI-based method of quantitatively assessing hepatic steatosis and is available as an option on MRI scanners made by several manufacturers [80,81].

Elastography techniques are recent developments and have not been widely validated in NAFLD. In particular, few patients have participated in studies of the use of the XL probe and 2D-SWE for the prediction of fibrosis. In addition, there is no consensus regarding the use of these elastography techniques in clinical practice in place of liver biopsy. Nevertheless, VCTE and MRE appear to be best suited for the evaluation of liver fibrosis in patients with NAFLD. In fact, several clinical algorithms for the diagnosis and monitoring of patients with NAFLD using VCTE and MRE have been proposed [82,83,84]. In addition, the combination of VCTE and the fibrosis-4 index or the NAFLD fibrosis score has been proposed to assess liver fibrosis [85,86,87]. Additional prospective randomized controlled trials are needed to compare the diagnostic and prognostic accuracy and the cost-effectiveness of these elastography techniques.

## Figures and Tables

**Figure 1 ijms-21-04039-f001:**
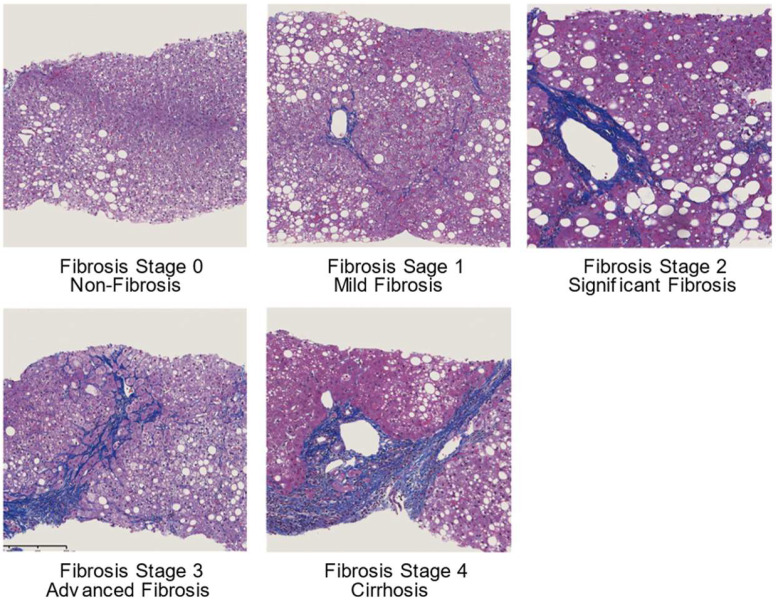
Liver fibrosis score in non-alcoholic fatty liver disease (Masson trichrome staining). Stage 1: Pericellular and perisinusoidal fibrosis in zone 3. Stage 2: Pericellular and perisinusoidal fibrosis with periportal fibrosis. Stage 3: Bridging fibrosis. Stage 4: Cirrhosis.

**Figure 2 ijms-21-04039-f002:**
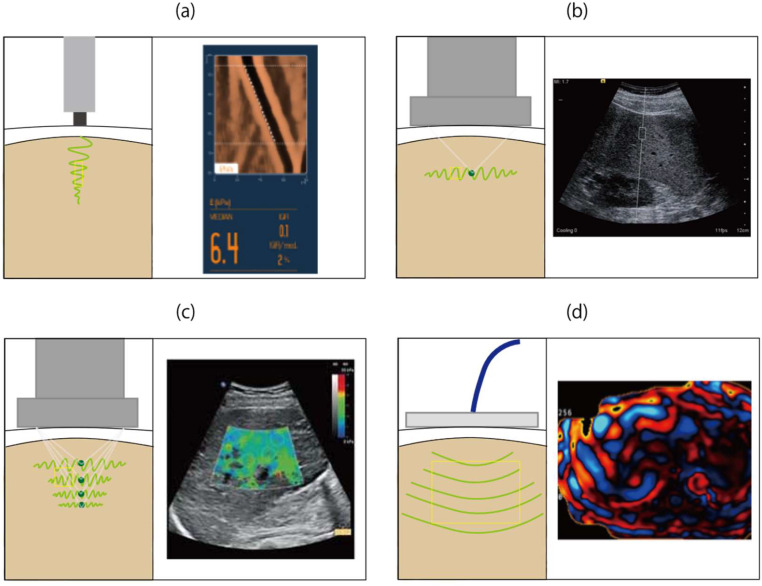
Illustrations and images of each elastography technique. (**a**) Vibration-controlled transient elastography transmits a mechanical pulse from the probe to the liver. (**b**) Point shear wave elastography and (**c**) two-dimensional shear wave elastography use acoustic radiation force impulse (green dots) to induce shear waves in liver tissue. (**d**) Magnetic resonance elastography uses a driver system to generate and transmit longitudinal waves into the liver. The green-yellow wave lines (**a**, **b**, **c**) show shear waves, and the curved lines (**d**) show vibrations in the liver. The yellow rectangles indicate the interrogated liver volume.

**Table 1 ijms-21-04039-t001:** Meta-analyses of elastography techniques for the diagnosis of liver fibrosis in patients with non-alcoholic fatty liver disease.

Technique	Author	Year	Reference No.	No. of Patients	No. of Studies					
VCTE M probe	Kwok et al.	2014	26	854	8	Fibrosis Stage	Range			
							Cut-off (kPa)	Sensitivity	Specificity	AUROC
						≥2	6.65–7.7	0.67–0.94	0.61–0.84	0.79–0.87
						≥3	8.0–10.4	0.65–1.00	0.75–0.97	0.76–0.98
						≥4	10.3–17.5	0.78–1.00	0.82–0.98	0.91–0.99
						Fibrosis Stage	Summary			
								Sensitivity	Specificity	
						≥2		0.79	0.75	
						≥3		0.85	0.85	
						≥4		0.92	0.92	
	Xiao et al.	2017	28	2495	16	Fibrosis Stage	Range			
							Cut-off (kPa)	Sensitivity	Specificity	
						≥2	5.8	0.90–0.94	0.42–0.80	
							6.65–7	0.58–1.00	0.45–0.84	
							7.25–11	0.53–0.84	0.70–0.93	
						≥3	6.95–7.25	0.67–0.70	0.65–0.68	
							7.6–8	0.65–1.00	0.66–0.90	
							8.7–9	0.76–0.88	0.63–0.88	
							9.6–11.4	0.69–1.00	0.84–0.97	
						≥4	7.9–8.4	0.93–1.00	0.76–0.79	
							10.3–11.3	0.78–1.00	0.82–0.90	
							11.5–11.95	0.69–0.90	0.85–0.91	
							13.4–22.3	0.41–1.00	0.76–0.98	
						Fibrosis Stage	Summary			
										AUROC (95% CI)
						≥2				0.83 (0.79–0.86)
						≥3				0.87 (0.83–0.90)
						≥4				0.92 (0.90–0.94)
	Jiang et al.	2018	27	1753	11	Fibrosis Stage	Range			
							Cut-off (kPa)	Sensitivity	Specificity	AUROC
						≥2	6.7–11.0	0.60–0.94	0.61–1.00	0.79–0.88
						≥3	8.0–12.5	0.57–1.00	0.76–0.97	0.76–0.99
						≥4	10.4–17.5	0.65–1.00	0.76–0.98	0.87–0.99
						Fibrosis Stage	Summary			
								Sensitivity (95% CI)	Specificity (95% CI)	AUROC (95% CI)
						≥2		0.77 (0.70–0.84)	0.80 (0.74–0.84)	0.85 (0.82–0.88)
						≥3		0.79 (0.69–0.87)	0.89 (0.84–0.92)	0.92 (0.89–0.94)
						≥4		0.90 (0.73–0.97)	0.91 (0.87–0.94)	0.94 (0.93–0.97)
VCTE XL probe	Xiao et al.	2017	28	318	3	Fibrosis Stage	Range			
							Cut-off (kPa)	Sensitivity	Specificity	
						≥2	4.8–8.2	0.57–0.92	0.37–0.90	
						≥3	5.7–9.3	0.57–0.91	0.54–0.90	
						≥4	7.2–16	0.71–1.00	0.70–0.91	
						Fibrosis Stage	Summary			
										AUROC (95% CI)
						≥2				0.82 (0.75–0.89)
						≥3				0.86 (0.78–0.94)
						≥4				0.94 (0.88–0.99)
pSWE	Liu et al.	2015	34	723	7	Fibrosis Stage	Range			
							Cut-off (m/s)	Sensitivity	Specificity	
						≥2	1.165–1.79	0.71–0.90	0.67–0.90	
						≥3	1.45–2.20	0.75–1.00	0.68–0.95	
						≥4	1.61–2.90	0.74–1.00	0.67–0.96	
						Fibrosis Stage	Summary			
								Sensitivity (95% CI)	Specificity (95% CI)	AUROC (95% CI)
						≥2		0.80 (0.76–0.84)	0.85 (0.81–0.89)	0.90
	Jiang et al.	2018	27	982	9	Fibrosis Stage	Range			
							Cut-off (m/s)	Sensitivity	Specificity	AUROC
						≥2	1.16–1.32	0.56–0.85	0.78–0.91	0.71–0.94
						≥3	1.34–1.77	0.59–1.00	0.74–0.96	0.76–0.99
						≥4	1.40–2.48	0.44–1.00	0.74–1.00	0.84–0.98
						Fibrosis Stage	Summary			
								Sensitivity (95% CI)	Specificity (95% CI)	AUROC (95% CI)
						≥2		0.70 (0.59–0.79)	0.84 (0.79–0.88)	0.86 (0.83–0.89)
						≥3		0.89 (0.73–0.96)	0.88 (0.82–0.92)	0.94 (0.91–0.95)
						≥4		0.89 (0.60–0.98)	0.91 (0.82–0.95)	0.95 (0.93–0.97)
	Lin et al.	2020	35	1147	13	Fibrosis Stage	Summary			
							Cut-off (m/s)	Sensitivity (95% CI)	Specificity (95% CI)	AUROC (95% CI)
						≥2	1.3	0.85	0.83	0.89 (0.85–0.91)
						≥3	2.06	0.9	0.9	0.94 (0.91–0.96)
						≥4	1.89	0.9	0.95	0.94 (0.92–0.95)
MRE	Singh et al.	2016	58	232	9	Fibrosis Stage	Summary			
							Cut-off (kPa)	Sensitivity (95% CI)	Specificity (95% CI)	AUROC (95% CI)
						≥1	2.88	0.75 (0.68–0.87)	0.77 (0.65–0.88)	0.86 (0.82–0.90)
						≥2	3.54	0.79 (0.76–0.90)	0.81 (0.72–0.91)	0.87 (0.82–0.93)
						≥3	3.77	0.83 (0.53–0.90)	0.86 (0.81–0.96)	0.90 (0.84–0.94)
						≥4	4.09	0.88 (0.82–1.00)	0.87 (0.77–0.97)	0.91 (0.76–0.95)
	Xiao et al.	2017	28	628	5	Fibrosis Stage	Range			
							Cut-off (kPa)	Sensitivity	Specificity	
						≥2	3.4–3.62	65.7–97.3	85.0–95.7	
						≥3	3.62–4.8	74.5–92.2	86.9–93.3	
						≥4	4.15–6.7	80.0–90.9	91.4–94.5	
						Fibrosis Stage	Summary			
										AUROC (95% CI)
						≥2				0.88 (0.83–0.92)
						≥3				0.93 (0.90–0.97)
						≥4				0.92 (0.80–1.00)

For the diagnosis of mild fibrosis (stage 1), significant fibrosis (stage 2), advanced fibrosis (stage 3), and cirrhosis (stage 4), histopathology was used as the reference standard. VCTE, vibration-controlled transient elastography; pSWE, point shear wave elastography; MRE, magnetic resonance elastography; AUROC, area under the receiver operating characteristic curve; CI, confidence interval.

**Table 2 ijms-21-04039-t002:** Advantages and limitations of elastography techniques.

	US-Based				MR-Based
	VCTE		pSWE	2D-SWE	MRE
	M Probe	XL Probe			
Confounder	Obesity		Obesity	Obesity	
	Inflammation	Inflammation	Inflammation	Inflammation	
	Ascites	Ascites			
					Iron Overload
	Cholestasis, Hepatic Congestion				
Sampling Volume of Liver	Little				Large
Technical Failure	6.7–29.2%		~2%	~5%	~5%
Cost	Low	Low	Moderate	Moderate	High
Availability	Good				Limited
HCC Screening	Blind technique		US Exam	US Exam	MRI Exam
Evaluation of Liver Fat Accumulation	CAP		-	-	PDFF
Guideline Recommendation	AASLD, EASL		-	-	AASLD

US, ultrasonography; MR, magnetic resonance; VCTE, vibration-controlled transient elastography; pSWE, point shear wave elastography; 2D-SWE, two-dimensional shear wave elastography; MRE, magnetic resonance elastography; HCC, hepatocellular carcinoma; MRI, magnetic resonance imaging; CAP, controlled attenuation parameter; PDFF, proton density fat fraction; AASLD, American Association for the Study of Liver Diseases; EASL, European Association for the Study of the Liver.

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
