# Peer review of "Elastography Techniques for the Assessment of Liver Fibrosis in Non-Alcoholic Fatty Liver Disease"

_ijms, 2020, doi:10.3390/ijms21114039_

Round 1

Reviewer 1 Report

Drs. Honda Y, et al reviewed various elastographic techniques for liver fibrosis assessment in NAFLD. The theme is very interesting and the description of the text is very well done. This manuscript provides useful information on the advantages and disadvantages of each method based on several meta-analysis data reported to date. However, since this manuscript is a review focusing on a very limited topic of the role of elastography in NAFLD, it would be more useful to summarize the meaning of results of each important original study and show it to readers, rather than just presenting the results of meta-analysis. Since each meta-analysis on the same subject is an analysis result including duplicate original studies, it is thought that the meaning of each meta-analysis study is reduced.

Author Response

Reviewer #1

Thank you for taking the time to review our manuscript, “Elastography Techniques for the Assessment of Liver Fibrosis in Non-Alcoholic Fatty Liver Disease” (ijms-804632), and for providing helpful comments concerning our study.

  1. Honda Y, et al reviewed various elastographic techniques for liver fibrosis assessment in NAFLD. The theme is very interesting and the description of the text is very well done. This manuscript provides useful information on the advantages and disadvantages of each method based on several meta-analysis data reported to date. However, since this manuscript is a review focusing on a very limited topic of the role of elastography in NAFLD, it would be more useful to summarize the meaning of results of each important original study and show it to readers, rather than just presenting the results of meta-analysis. Since each meta-analysis on the same subject is an analysis result including duplicate original studies, it is thought that the meaning of each meta-analysis study is reduced.

Thank you for raising an important point. In response, we have created Supplementary Table 1. We have added additional material to the Introduction, as shown below.

Lines 48-49: “Supplementary Table 1 lists 47 studies included in the meta-analyses.”

Reviewer 2 Report

This manuscript is focused on the advantages and limitations of several elastgraphy techinques. I think that this is beneficial for many readers. Especially, Table 2 is very good. But there are some comments.

  1. In Table.1, >=2, >=3, etc are fibrosis score? Please describe it in Table 1.
  2. liver gibrosis score are not famililar for general physicians. Please represent new figure about the relationship between histology and liver fibrosis score.
  3. There are too many coauthors in this review, although this is a concise review. The authors should select 4 authors.
  4. I think that other noninvasive methods of serological markers such as Fib-4 index are also promising. Does combination of elastgraphy and other methods (CT or serological markers) improve the accuracy for diagnosis? If there are clinical reports about them, please describe it.

Author Response

Reviewer #2

Thank you for taking the time to review our manuscript, “Elastography Techniques for the Assessment of Liver Fibrosis in Non-Alcoholic Fatty Liver Disease” (ijms-804632), and for providing helpful comments concerning our study.

  1. In Table.1, >=2, >=3, etc are fibrosis score? Please describe it in Table 1.

Thank you very much for your comment. We have revised Table 1.

  1. Liver gibrosis score are not famililar for general physicians. Please represent new figure about the relationship between histology and liver fibrosis score.

Thank you very much for suggesting this addition. We have created a new figure (Figure 1) showing the relationship between the histology and the liver fibrosis score. We have added additional material to the Introduction, as shown below.

Lines 28–30: “Liver fibrosis is categorized into non-fibrosis (stage 0), mild fibrosis (stage 1), significant fibrosis (stage 2), advanced fibrosis (stage 3), and cirrhosis (stage 4) (Figure 1) [9].”

  1. There are too many coauthors in this review, although this is a concise review. The authors should select 4 authors.

Thank you very much for your comment. We have selected four authors.

  1. I think that other noninvasive methods of serological markers such as Fib-4 index are also promising. Does combination of elastgraphy and other methods (CT or serological markers) improve the accuracy for diagnosis? If there are clinical reports about them, please describe it.

Thank you for raising this important point. We have added further details regarding this issue, as shown below.

Lines 298-299: “In addition, the combination of VCTE and the fibrosis-4 index or the NAFLD fibrosis score has been proposed to assess liver fibrosis [85-87].”

Round 2

Reviewer 1 Report

Although the part I mentioned in the 1st review has not been completely revised, it seems that the authors have made great efforts to make up for the above-mentioned parts.